# Quinine Esters with 1,2-Azole, Pyridine and Adamantane Fragments

**DOI:** 10.3390/molecules27113476

**Published:** 2022-05-27

**Authors:** Gulim K. Mukusheva, Aigerym R. Zhasymbekova, Roza B. Seidakhmetova, Oralgazy A. Nurkenov, Ekaterina A. Akishina, Sergey K. Petkevich, Evgenij A. Dikusar, Vladimir I. Potkin

**Affiliations:** 1Karaganda Buketov University, Karaganda 100024, Kazakhstan; mukusheva1977@list.ru (G.K.M.); aigera-93-93@mail.ru (A.R.Z.); rozabat@mail.ru (R.B.S.); 2Institute of Organic Synthesis and Coal Chemistry of the Republic of Kazakhstan, Karaganda 100008, Kazakhstan; nurkenov_oral@mail.ru; 3Institute of Physical Organic Chemistry, National Academy of Sciences of Belarus, 220072 Minsk, Belarus; peetsk7777@gmail.com (S.K.P.); dikusar@ifoch.bas-net.by (E.A.D.); potkin@ifoch.bas-net.by (V.I.P.)

**Keywords:** quinine, esters, isoxazole, isothiazole, pyridine, adamantane, quaternary pyridinium salts, antiviral activity, antimicrobial activity, analgesic activity

## Abstract

An efficient method of producing quinine derivatives via reaction of acylation with 4,5-dichloroisothiazole-3-, 5-arylisoxazole-3-, adamantane- and hydrochlorides of pyridine-3- and pyridine-4-carbonyl chlorides was developed. All synthesized compounds were tested for antiviral, antimicrobial and analgesic activity. The most pronounced antibacterial activity was shown by the compounds **2e**, **3b**, **3c** and **3e** with isoxazole and pyridine fragments. It was found that most of the tested compounds showed significant analgesic activity reducing the pain response of animals to the irritating effect of acetic acid.

## 1. Introduction

Quinine is the main alkaloid of the bark of various cinchona (*Cinchona*) species, which has a pronounced activity against malarial plasmodia and made it possible to use quinine as an effective treatment for malaria for a long time [1]. However, quinine at the same time has a toxic side effect [2,3], causing tinnitus, hearing and vision impairment, dizziness, heart palpitations, vomiting, hypotension, hypoglycemia and kidney failure. This was the reason for replacing quinine with more effective and safer antimalarial drugs, such as artemisinin [4].

Along with its antimalarial activity, quinine has been reported to possess antipyretic, antibacterial and antifungal properties [5,6,7]. Currently, quinine and its derivatives are considered as potential treatments for COVID-19 [8,9,10,11]. Taking into account the high potential activity, numerous studies are underway in the world aimed at searching for new drugs among quinine derivatives, including the esterification reaction of the C-9 hydroxyl group [12,13,14,15]. 

Nowadays, antimicrobial resistance is considered one of the greatest problems facing humans, because many bacterial strains have become resistant to available antibiotics. Thus, the discovery of new effective antimicrobial agents, especially from traditional medicinal plants and their derivatives, is urgently needed. Moreover, a screening study of some quinine esters suggested that they have moderate antimicrobial activity against human pathogenic bacteria strains *Escherichia coli*, *Staphylococcus aureus*, *Pseudomonas aeruginosa* and *Bacillus substillis*; thus, the compounds potentially work as antimicrobial agents for both Gram-negative and Gram-positive bacterial strains [12]. In this present study, we synthesized several derivative compounds of quinine with specific functional groups in order to understand how different functional groups serve different antimicrobial actions. 

Pyridine, isoxazole and isothiazole heterocycles are widely used structural units in the design and synthesis of new biologically active compounds and are included in the molecular structures of a large number of bioactive substances used in medical practice [16,17,18,19,20,21]. The combination of alkaloid, pyridine and 1,2-azole fragments in one molecule can improve the activity and impart new useful properties to target compounds, and the high lipophilicity with the bulk structure of the adamantane radical can significantly promote and modify the pharmacological action of various biologically active compounds by creating favorable conditions for their transport through biological membranes [22].

Promising data on the antiviral activity of quinine from numerous literary sources [8,9,10,11,23,24] prompted us to study the antiviral properties of obtained derivatives. We also examined the analgesic and fungicidal activity of the target compounds.

The use of an ester bond for the covalent bonding of various pharmacophore groups and structural fragments of natural as well as synthetic origin can serve as a prime example of molecular design. Drugs with an ester group often perform prodrug functions. In the body, “prodrugs” undergo biotransformation and turn into the true drug. Prodrugs are common tools for overcoming drawbacks typically associated with drug formulation and delivery, with ester prodrugs providing a classic strategy for masking polar alcohol and carboxylic acid functionalities and improving cell permeability [25,26]. The ester group in a drug molecule can also perform the following functions: (a) be part of the pharmacophore—a grouping of atoms that determines the pharmacological effect of the substance; (b) protect reactive groups from exposure to adverse environmental factors (air oxygen, light) and the internal environment of the body (enzymes); (c) reduce the toxicity and irritant properties of the substance; (d) improve the pharmacokinetic properties of the substance. An additional factor that contributes to the popularity of esters is that the synthesis of an ester is often straightforward. 

## 2. Results and Discussion

### 2.1. Chemistry

In the framework of this work, quinine esters with heterocyclic 1,2-azole, pyridine and adamantane fragments were obtained. It should be noted that quinine esters with isoxazole and isothiazole fragments have not been previously obtained; however, there are methods for the synthesis of quinine derivatives with a pyridine heterocycle [14,15], and some of them have shown high insecticidal activity against *Mythimna separata*. The article [14] presents a one-step synthesis method using DCC and DMAP as a catalyst, and the yield of esters of nicotinic and isonicotinic acids and quinine was 44 and 24%, respectively. 

The method proposed by us is based on the reaction of quinine acylation with hydrochlorides of pyridine-3- and pyridine-4-carbonyl chlorides. It can compete with that stated in [14] due to higher yields (>80%) and the absence of the need for further purification of the products isolated from the reaction mixture. Esters **2a**–**f** were obtained by acylation of quinine **1** with various acid chlorides in dichloromethane in the presence of triethylamine at room temperature in 86–91% yield (Figure 1). 

Based on the synthesized derivatives **2a**–**f**, quaternary pyridinium salts (iodomethylates) were obtained. Quaternization proceeded with the participation of quinoline, quinuclidine and pyridine nitrogen atoms and led to the formation of diiodomethylates **3a**–**c** in 94–98% yield and triiodomethylates **3d**,**e** in 83–85% yield. The quaternization reaction of the obtained esters proceeds completely with a 4-fold excess of the alkylating agent, and the resulting salts precipitate out of the solution. 

Quaternization of quinine esters makes it possible to increase the water solubility of compounds, which is important for choosing the most rational ways of introducing drugs into the body. Pyridinium salts are also known to inhibit the growth of various microorganisms such as bacteria, viruses, and fungi. Thus, quinine alkylation displayed an increase in antibacterial activity of obtained amphiphiles against a number of Gram-positive and Gram-negative strains [27]. 

The obtained compounds were identified on the basis of IR, UV, mass and NMR spectra (^1^H and ^13^C), as well as elemental analysis (please see the Appendix A).

All synthesized compounds were tested for antiviral, antimicrobial and analgesic activity.

### 2.2. Evaluation of the Biological Activity

#### 2.2.1. Antiviral Activity

Cytotoxicity and chicken embryo lethality of test compounds were studied. 

The acute toxicity of the samples was assessed at various doses in various in vitro models and in a 10-day-old chick embryo model. The study of acute toxicity of quinine derivatives “in vitro” was carried out on macrophage models of outbred mice. The interval of the dose range was determined, first of all, by the interval of acceptable values for the number of compounds used in further studies for antiviral activity. The analysis of acute cytotoxicity of compounds “in vitro” was carried out in the dose range of 0.003–0.4 mg, corresponding to effective doses of compounds with antiviral properties. Cytotoxicity of substances was determined by studying the effect of various doses of compounds on cell viability, by the method of dehydrogenase activity detection (MTT test).

It was found that in the tested dose range, all the studied compounds did not reach the LD50. At the maximum dose of 0.4 mg/chick embryo, the toxicity (LD50) of the test compounds was not manifested; therefore, further presence study of the antiviral activity was carried out in the dose range from 0.4 mg/chick embryo or less. Thus, in the determination of acute toxicity “in vitro” and on the model of 10-day-old chicken embryos, the studied compounds did not reveal toxic properties at the maximum of the tested doses.

The virus-inhibiting activity of the samples was studied.

The virus-inhibiting activity of the preparations was studied at concentrations from 0.0016% to 0.2%, which corresponded to doses of 0.003–0.4 mg per chick embryo (0.06–8 mg/kg). It was found that in the given dose range, the studied compounds suppress the reproduction of the human influenza A/Almaty/8/98 (H3N2) virus by no more than 12%. At the same time, compounds **2e**, **3b**, **3c**, **3d** and **3e** showed more pronounced antiviral properties in relation to the human influenza virus strain A/Almaty/8/98 (Table 1) than other compounds.

The antiviral activity of the compounds was also studied on the model of avian influenza virus strain A/FPV/34/1 (H7N1). It was shown that **2b** and **2f** did not exhibit virus-inhibiting properties at all doses studied. Compounds **2e**, **3b**, **3d** and **3e** at a dose of 0.4 mg/chick embryo suppressed the reproduction of the influenza virus strain A/FPV/34/1 (H7N1) up to 16.5% (Table 2).

Further study of the antiviral activity of preparations **2b**, **2e**, **2f** and **3b** was carried out on the model of swine influenza virus strain A/swine/Iowa/30 (H1N1). It was found that at a dose of more than 0.08 mg/embryo, all the studied compounds showed mild antiviral properties. With the increase in the dose to 0.4 mg/chick embryo, the antiviral activity of the compounds **2e**, **3b**, **3c** and **3e** increased to 30% (Table 3).

The virucidal activity of the samples was studied in the dose range 0.003–0.4 mg/chick embryo against influenza virus strains A/Almaty/8/98 (H3N2), A/FPV/34/1 (H7N1) and A/swine/Iowa/30 (H1N1).

On three strains of the influenza virus, it was found that the compounds **2b**, **2e**, **2f** and **3b** do not have virucidal properties.

Thus, it was shown that the majority of the tested compounds did not show pronounced antiviral properties in the range of the studied doses. However, it was found that in a number of compounds under study, samples **2e**, **3b**, **3c** and **3e** with isoxazole and pyridine-4- fragments have more pronounced antiviral properties and exceed the activity of the reference drug Amizon in their action. Derivatives with an isonicotinic fragment were found to have a higher antiviral activity against all studied strains (Table 1, Table 2 and Table 3).

#### 2.2.2. Antimicrobial Activity

According to the literature data, it has been established that quinine derivatives have a pharmacological effect against *Staphylococcus aureus*, *Streptococcus pneumoniae*, *Pseudomonas aeruginosa* and *Escherichia coli* [28]. It has been proven that quinoline derivatives inhibit DNA synthesis, promoting the cleavage of bacterial DNA gyrase, resulting in the death of bacterial cells [29].

The ability of compounds of the quinoline series, such as β-lactam antibiotics used to prevent pathogenic processes in the body, caused in particular by *Staphylococcus aureus*, to participate in the irreversible inhibition of the activity of transpeptidase, a penicillin-binding protein that catalyzes the formation of peptidoglycan, an essential component of the bacterial cell wall, is described and leads to the death of the pathogen.

The above data indicate the need to study the antimicrobial properties of the new derivatives of quinoline compounds synthesized by us in order to identify potential antimicrobial agents.

As a result of the antimicrobial study, it was found that the test compounds exhibit antibacterial activity in varying degrees of severity against the presented opportunistic test strains (Table 4).

Analysis of the antimicrobial activity of the test substances showed that its manifestation depends on the type of pathogenic microorganism.

Test strain *Staphylococcus aureus* is the most sensitive to all the presented compounds (except **2a**, **2c**), for which their minimum inhibitory concentration is between 7 and 63 µM.

The most pronounced antibacterial activity was shown by the compounds **2e**, **3b**, **3c** and **3e** against the Gram-positive test strain of *Staphylococcus aureus ATCC 6538*. The antibacterial effect of these compounds against this test strain reached 7 μM, even better than that of ceftriaxone. 

The obtained data on antimicrobial activity against *Staphylococcus aureus* allow us to conclude that quaternization of quinine esters increases antibacterial activity compared to the initial substrates (Figure 1).

For test strains of *Bacillus subtilis* and *Escherichia coli*, the minimum inhibitory concentration of a number of test compounds is in the range of 14.6–116.4 µM. At the same time, the antibacterial activity of these compounds against *Bacillus subtilis* and *Escherichia coli* is lower than that of the reference drug.

The Gram-negative test strain of *Pseudomonas aeruginosa* turned out to be the most resistant to the action of these compounds. None of the test compounds showed antibacterial activity against this microorganism.

Antifungal activity was revealed in the compounds **2f** and **3d**. Antifungal activity against the yeast-like fungus *Candida albicans*, which causes human opportunistic fungal infections, was observed in these substances at concentrations of 58.5 and 102.7 μM.

Thus, in the series of new synthesized derivatives of quinine, compounds with antibacterial activity comparable to the activity of the drug ceftriaxone were identified. This allows us to consider this series of compounds as very promising for the search for new potential antibacterial drugs, which requires further in-depth studies.

#### 2.2.3. Analgesic Activity In Vivo

In the course of studying the analgesic activity induced by novel compounds, the animals were observed from the moment of modeling the vinegar writhing. It was found that most of the test compounds, when administered once at a dose of 25 mg/kg 1 h before the stimulus, significantly reduce the pain response of animals to the irritating effect of acetic acid (Table 5). 

The greatest analgesic effect among the studied contaminated forms and potential pharmaceutical substances of **2b**, **2d**, **2e**, **2f**, **3c** and **3e** caused a significant decrease in the amount of vinegar writhing in mice by 47.6, 41.4, 49.4, 48.8, 46.1 and 52.7% respectively. The analgesic activity level of these compounds is comparable to sodium diclofenac.

Compounds **2b**, **2d**, **2e**, **2f**, **3c** and **3e** at doses of 25 mg/kg showed analgesic activity in the model of chemical irritation of the peritoneum (test “vinegar cramps”), showing a significant decrease in the pain response of visceral nociceptors to the irritating effect of acetic acid in comparison with the control.

## 3. Materials and Methods

### 3.1. General Chemistry Section 

UV spectra were recorded on a Varian Cary 300 spectrophotometer using quartz cuvettes with *l* = 1 cm. The concentration of the studied compounds in methanol was 4–6 × 10^−5^ mol/L. IR spectra were registered on a Thermo Nicolet Protege 460 Fourier transform spectrometer in KBr pellets. 

^1^H and ^13^C NMR spectra were acquired on a Bruker Avance 500 spectrometer (500 and 125 MHz, respectively) in DMSO-*d*6 and CDCl_3_. The residual solvent signals (DMSO-*d*6, δH 2.5, δC 40.1 ppm; CDCl_3_, δH 7.26, δC 77.2 ppm) were used as internal standard. The assignment of signals in the ^13^C NMR spectra was performed using the DEPT technique. 

Liquid chromatography–mass spectrometry spectra were recorded on an Agilent 1200 LC-MS system with an Agilent 6410 Triple Quad Mass Selective Detector with electrospray ionization in the positive ion registration mode (MS2 scanning mode). An Agilent ZORBAX Eclipse XDB-C18 (4.6 × 50 mm, 1.8 μm) column was used. The mobile phase was MeCN–H_2_O + 0.05% HCO_2_H with gradient elution from 40 to 90% MeCN in 10 min. A flow rate of 0.5 mL/min was used. 

Elemental analysis was performed on a Vario MICRO cube CHNS-analyzer. The halogen content was determined by classical microanalysis using a modified Pregl’s method. Melting points were determined on a Kofler bench. 

Reagents and solvents used were of analytical grade with the content of the main component being more than 99.5%. Dichloromethane was preliminarily kept for 1 day over CaCl_2_ to remove 0.5% of ethanol used for stabilizing dichloromethane. 5-Arylisoxazole-3-carboxylic and 4,5-dichloroisothiazole-3-carboxylic acids and acid chlorides were synthesized according to previously described procedures [30,31].

### 3.2. In Vitro Biological Assays

#### 3.2.1. Antiviral Activity

The virus-inhibiting properties of the compounds were studied in experiments with orthomyxoviruses on chick embryos. Determination of antiviral properties was performed by the “screening test” method, designed to neutralize the virus in the amount of 100 EID50 given concentrations of the tested compounds. The presence of a difference in virus titer in comparison with the control was considered as a criterion for antiviral activity. In this case, as a rule, only complete suppression of the virus titer was taken into account.

The virucidal activity of the studied samples was determined by treating the virus-containing material with chemical compounds at 37 °C for 30 min, followed by titration of the infectivity of the treated material. The real virucidal effect was taken as the difference between the virus titer in the sample without exposure and its titer after. If the difference in titers was 1.0–2.0 lg, then the substance was considered to have moderate or pronounced antiviral activity [32]. The infectious titer of viruses was determined by the method of Reed and Muench [33].

The antiviral drug Amizon was used as a reference drug.

The following viruses were used in the work:

Orthomyxoviruses: avian influenza virus, strain A/FPV/34/1 (H7N1); human influenza virus, strain A/Almaty/8/98 (H3N2); and swine influenza virus A/swine/Iowa/30 (H1N1) were obtained from the collection of microorganisms at the Research Institute for Biological Safety Problems (Kazakhstan). Viruses were grown in the allantoic cavity of 10-day-old chicken embryos for 36 h at 37 °C.

The hemagglutinating activity of viruses was determined according to the standard method [34] using a 0.75% suspension of chicken erythrocytes.

Phosphate buffered saline (PBS, pH 7.4) was purchased from Amresco (Solon, OH, USA). Ten-day-old chicken eggs and 50% chicken red blood cell (cRBC) suspensions were obtained from the Almaty chicken factory farm (Almaty, Kazakhstan). 

To assess the toxicity of the obtained test compounds, the MTT cytotoxicity test using macrophages of outbred mice was used. The technique is used to evaluate the cytotoxicity of solutions of compounds in the experiment and is based on the ability of living cell dehydrogenases to reduce unstained forms of 3–4,5-dimethylthiazol-2-yl-2,5-diphenylterarazole (MTT reagent) to blue crystalline formazan, soluble in dimethyl sulfoxide.

Acute toxicity was determined in a 10-day-old chick embryo model. The toxicity of the test materials was determined by inoculating 0.2 mL of the compound into the chorioallantoic cavity of chicken embryos (embryotoxicity). The toxicity of the preparations was determined by the death of chicken embryos within 4 days after the inoculation of materials.

Standard methods for finding the mean values and their mean errors were used for the statistical processing of the results.

#### 3.2.2. Antimicrobial Activity

The antimicrobial activity of the samples was studied on the reference test micro-organisms recommended by the State Pharmacopoeia of the Republic of Kazakhstan: facultative anaerobic Gram-positive cocci *Staphylococcus aureus ATCC 6538*, aerobic Gram-positive spore-forming rods *Bacillus subtilis ATCC 6633*, Gram-negative facultative anaerobe rods *Escherichia coli ATCC 25922*, aerobic *Pseudomonas aeruginosa ATCC 27853* and yeast fungus *Candida albicans ATCC 10231* by the method of random dilutions with the determination of the minimum inhibitory concentration (MIC) [35,36]. The test strains of microorganisms used in the study were obtained from the American Type Culture Collection [29].

The antibacterial drug ceftriaxone and the antifungal drug nystatin were used as reference drugs.

The minimum inhibitory concentration (MIC) of the samples was determined by the method of serial dilution of ethanol solutions of the test samples in nutrient broth. For serial dilutions, suspensions of test strains at the concentration of 106 CFU/mL were used. The suspension of test strains of microorganisms was prepared from daily cultures grown on slant agar at the temperature of 37 °C for 24 h; for the yeast fungus *Candida albicans*, cultures were grown at 30 °C for 48 h.

Then, the test tube containing a clear suspension and the lowest concentration of the antimicrobial agent was selected by visual determination of turbidity in each test tube. This concentration was in line with the MIC. The results were averaged over the data of three experiments.

### 3.3. Analgesic Activity In Vivo

The experimental part was carried out in accordance with the “Rules of the European Convention for the Protection of Vertebrate Animals used for Experimental and Other Scientific Purposes” and in accordance with the requirements for the study of new pharmacological substances [37]. The analgesic effect of the synthesized compounds was carried out using chemical stimulus on outbred white mice with weight in the range of 23 to 35 g. The experimental animals were kept in standard vivarium conditions on a normal diet. Five groups containing six animals each were formed (control, reference drug “diclofenac sodium”, three novel substances).

The analgesic effect of the samples was evaluated in the chemical irritation test of the peritoneum (test “vinegar cramps”). The abdominal constriction test is a visceral inflammatory pain model (acute peritonitis model). When visceral receptors are irritated with acetic acid, abdominal muscle contraction, hind limb extension and body elongation are observed [38]. A 0.75% solution of acetic acid was injected intraperitoneally in an amount of 0.1 mL per 10 g of animal weight. The potential pharmaceutically active substances were injected intragastrically at a dose of 25 mg/kg 30 min before the administration of the acetic acid. Immediately after the introduction of the stimulus, the latent time of the onset of the pain reaction “writhing” was recorded, and the writhings were counted for 30 min. The analgesic effect of compounds was determined by the ability to reduce the number of “writhings” counted for 10, 15, 20 and 30 min, compared with the corresponding indicators in the control animal group. The model drug was the non-steroidal anti-inflammatory drug diclofenac sodium, which was tested at an effective dose of 8 mg/kg (ED50 = 8 mg/kg). Control animals received the equivalent volume of starchy mucus.

Analgesic activity was expressed as a percentage reduction in the number of acetic writhings in experimental rats compared to controls.

Statistical processing was carried out by parametric statistical methods with the calculation of the arithmetic mean and standard error. Differences were considered significant at the achieved significance level *p* < 0.05.

The analgesic activity level of this compound is comparable to sodium diclofenac.

### 3.4. Experimental Section

#### 3.4.1. General Procedure for the Synthesis of Compounds **2a**–**f**

Quinine (2.6 g, 0.008 mol) was dissolved in 100 mL of dry dichloromethane. Then, 1.0 g (0.01 mol) of triethylamine and 0.009 mol of 1,2-azole-3- (**2a**–**c**) or adamantane- (**2f**) carbonyl chlorides were successively added to the resulting solution under stirring. The mixture was stirred for 1 h and left for 15 h at 20–23 °C. The mixture was washed with water (2 × 200 mL, 1 h stirring) and 5% sodium bicarbonate solution (2 × 200 mL, 1 h stirring). The organic layer was separated and dried over anhydrous Na_2_SO_4_. The solvent was removed, and the residue was crystallized from a mixture of ether and hexane (1:1).

The procedure for the synthesis of quinine esters with a pyridine fragment (**2d**,**e**) is similar to the previous one, except for the amount of the used triethylamine. Here, 2.6 g (0.008 mol) of quinine, 2.4 g (0.024 mol) of Et_3_N and 1.6 g (0.009) mol of hydrochloride of nicotinic or isonicotinic carbonyl chlorides were taken into the reaction.

*(R)-(6-Methoxyquinolin-4-yl)[(1S,2R,4S,5R)-5-vinylquinuclidin-2-yl]methyl 4,5-dichloroisothiazole-3-carboxylate* (**2a**): white solid; yield 88%; mp 131–132 °C; UV (MeOH c = 4 × 10^−5^ M) λ_max_ (log ε) 232 (4.57), 274 (3.90), 317 (3.70), 333 (3.70) nm; IR (KBr) ν 3075, 3010 (C=C–H); 2996, 2964, 2946, 2906, 2889, 2863 (C–H_aliph_); 1732 (C=O); 1623, 1589, 1560 1512 (C–C_arom_); 1464, 1455, 1403, 1354; 1263, 1213 (C–O); 1174, 1082, 1036, 975, 954, 906, 859, 843, 822 cm^−1^; ^1^H NMR (DMSO–*d*6, 500 MHz) δ 1.40–1.49 (1H, m, CH_2_), 1.55 (1H, dd, *J* = 12.4, 7.6 Hz, CH_2_), 1.70–1.77 (1H, m, CH_2_), 1.77–1.82 (1H, m, CH), 1.93–2.03 (1H, m, CH_2_), 2.17–2.26 (1H, m, CH–CH=CH_2_), 2.39–2.49 (2H, m, CH_2_ + CH_2_), 2.83 (1H, dd, *J* = 12.8, 10.5 Hz, CH_2_), 3.13–3.22 (1H, m, CH_2_), 3.50 (1H, q, *J* = 8.3 Hz, CH–N), 3.95 (3H, s, OMe), 4.95–5.07 (2H, m, CH=CH_2_), 5.89–6.00 (1H, m, CH=CH_2_), 6.59 (1H, d, *J* = 8.6 Hz, CH–O), 7.44 (1H_quin_, dd, *J* = 9.2, 2.6 Hz), 7.56 (1H_quin_, d, *J* = 4.5 Hz), 7.63 (1H_quin_, d, *J* = 2.6 Hz), 7.96 (1H_quin_, d, *J* = 9.2 Hz), 8.71 (1H_quin_, d, *J* = 4.5 Hz); ^13^C NMR (DMSO-*d*6, 125 MHz) δ 25.76 (CH_2_), 27.62 (CH), 27.74 (CH_2_), 40.36 (CH–CH=CH_2_), 42.21 (CH_2_), 56.10 (OMe), 56.24 (CH_2_), 60.05 (CH–N), 76.34 (CH–O), 102.75 (1CH_quin_), 114.91 (=CH_2_), 119.89 (1CH_quin_), 122.13 (1CH_quin_), 131.91 (1CH_quin_), 142.88 (CH=CH_2_), 148.13 (1CH_quin_), 125.16, 127.40, 144.19, 144.55, 151.26, 153.49, 157.90, 158.48 (C=O) (8C_quater_); MS *m*/*z* (*I*_rel_, %) 504.10 [M+H]^+^ (42.3); Anal. calcd. for C_24_H_23_Cl_2_N_3_O_3_S (504.43): C, 57.15; H, 4.60; Cl, 14.06; N, 8.33; S, 6.36%; Found: C, 57.44; H, 4.71; Cl, 13.88; N, 8.13; S, 6.22%.

*(R)-(6-methoxyquinolin-4-yl)[(1S,2R,4S,5R)-5-vinylquinuclidin-2-yl]methyl 5-phenylisoxazole-3-carboxylate* (**2b**): white solid; yield 89%; mp 158–159 °C; UV (MeOH c = 4 × 10^−5^ M) λ_max_ (log ε) 232 (4.56), 272 (4.22), 319 (3.58), 333 (3.67); IR (KBr) ν 3130 (CH_isox_), 3075, 3000 (C=C–H); 2945, 2924, 2882, 2860 (C–H_aliph_); 1736 (C=O), 1622, 1591, 1571, 1512 (C–C_arom_); 1461, 1446, 1320, 1297; 1264, 1242 (C–O); 1172, 1138, 1100, 1084, 1037, 1017, 957, 947, 931, 851, 821, 803, 781, 765, 688, 677 cm^−1^; ^1^H NMR (DMSO–*d*6, 500 MHz) δ 1.42–1.51 (1H, m, CH_2_), 1.60 (1H, dd, *J* = 12.7, 7.6 Hz, CH_2_), 1.70–1.78 (1H, m, CH_2_), 1.78–1.83 (1H, m, CH), 1.98–2.08 (1H, m, CH_2_), 2.19–2.27 (1H, m, CH–CH=CH_2_), 2.41–2.56 (2H, m, CH_2_ + CH_2_), 2.88 (1H, dd, *J* = 13.3, 10.2 Hz, CH_2_), 3.32–3.39 (1H, m, CH_2_), 3.50 (1H, q, *J* = 8.3 Hz, CH–N), 3.96 (3H, s, OMe), 4.96–5.07 (2H, m, =CH_2_), 5.93–6.02 (1H, m, CH=CH_2_), 6.62 (1H, d, *J* = 8.5 Hz, CH–O), 7.45 (1H_quin_, *J* = 9.2, 2.6 Hz), 7.54–7.57 (3H_Ar_, m), 7.57 (CH_isox_, s), 7.59 (1H_quin_, d, *J* = 4.5 Hz), 7.62 (1H_quin_, d, *J* = 2.6 Hz), 7.94–8.00 (3H, m, 1H_quin_+2H_Ar_), 8.72 (1H_quin_, d, J = 4.5 Hz); ^13^C NMR (DMSO-*d*6, 125 MHz) δ 25.44 (CH_2_), 27.69 (CH), 27.77 (CH_2_), 39.87 (CH–CH=CH_2_), 42.23 (CH_2_), 56.21 (OMe), 56.26 (CH_2_), 60.06 (CH–N), 76.36 (CH–O), 101.47 (CH_isox_), 102.70 (1CH_quin_), 114.94 (=CH_2_), 119.82 (1CH_quin_), 122.13 (1CH_quin_), 126.47 (2CH_Ar_), 129.92 (2CH_Ar_), 131.71 (1CH_quin_), 131.95 (1CH_Ar_), 142.94 (CH=CH_2_), 148.14 (1CH_quin_), 126.53, 127.28, 144.15, 144.58, 157.02, 157.92, 159.31, 172.06 (8C_quater_); MS *m*/*z* (I_rel_, %) 496.30 [M+H]^+^ (44.0); Anal. calcd. for C_30_H_29_N_3_O_4_ (495.58): C, 72.71; H, 5.90; N, 8.48%; Found: C, 73.09; H, 6.05; N, 8.35%.

*(R)-(6-methoxyquinolin-4-yl)[(1S,2R,4S,5R)-5-vinylquinuclidin-2-yl]methyl 5-(p-tolyl)isoxazole-3-carboxylate* (**2c**): white solid; yield 86%; mp 148–149 °C; UV (MeOH c = 6 × 10^−5^ M) λ_max_ (log ε) 233 (4.58), 278 (4.30), 317 (3.70), 332 (3.70); IR (KBr) ν 3137 (CH_isox_), 3072, 3029 (C=C–H); 2945, 2923, 2882, 2866 (C–H_aliph_); 1737 (C=O); 1622, 1592, 1513(C–C_arom_); 1460, 1446, 1316, 1295; 1265, 1240 (C–O); 1172, 1136, 1112, 1084, 1037, 1020, 999, 947, 928, 851, 823, 812, 800, 781, 715, 687, 677, 567, 501 cm^−1^; ^1^H NMR (DMSO–*d*6, 500 MHz) δ 1.41–1.50 (1H, m, CH_2_), 1.58 (1H, dd, *J* = 13.2, 7.6 Hz, CH_2_), 1.70–1.77 (1H, m, CH_2_), 1.78–1.84 (1H, m, CH), 1.98–2.07 (1H, m, CH_2_), 2.18–2.27 (1H, m, CH–CH=CH2), 2.37 (3H, s, Me), 2.40–2.49 (2H, m, CH_2_ + CH_2_), 2.85 (1H, dd, *J* = 13.5, 10.0 Hz, CH_2_), 3.15–3.24 (1H, m, CH_2_), 3.50 (1H, q, *J* = 8.3 Hz, CH–N), 3.96 (3H, s, OMe), 4.96–5.07 (2H, m, =CH_2_), 5.92–6.02 (1H, m, CH=CH_2_), 6.59 (1H, d, *J* = 8.4 Hz, CH–O), 7.37 (2H_Ar_, d, *J* = 8.1 Hz), 7.45 (1H_quin_, dd, *J* = 9.2, 2.6 Hz), 7.50 (CH_isox_, s), 7.58 (1H_quin_, d, J = 4.5 Hz), 7.62 (1H_quin_, d, *J* = 2.6 Hz), 7.85 (2H_Ar_, d, *J* = 8.1 Hz), 7.97 (1H_quin_, d, *J* = 9.2 Hz), 8.72 (1Hquin, d, *J* = 4.5 Hz); ^13^C NMR (DMSO-*d*6, 125 MHz) δ 21.59 (Me), 25.45 (CH_2_), 27.68 (CH), 27.76 (CH_2_), 39.88 (CH–CH=CH_2_), 42.22 (CH_2_), 56.19 (OMe), 56.26 (CH_2_), 60.07 (CH–N), 76.30 (CH–O), 100.79 (CH_isox_), 102.69 (1CH_quin_), 114.92 (=CH_2_), 119.82 (1CH_quin_), 122.12 (1CH_quin_), 126.40 (2CH_Ar_), 130.44 (2CH_Ar_), 131.94 (1CH_quin_), 142.94 (CH=CH_2_), 148.14 (1CH_quin_), 123.88, 127.88, 141.75, 144.17, 144.58, 156.94, 157.90, 159.35, 172.21 (9C_quater_); MS *m*/*z* (I_rel_, %) 510.20 [M + H]^+^ (52.7); Anal. calcd. for C_31_H_31_N_3_O_4_ (509.61): C, 73.06; H, 6.13; N, 8.25%; Found: C, 73.44; H, 6.36; N, 8.11%. 

*(R)-(6-methoxyquinolin-4-yl)[(1S,2R,4S,5R)-5-vinylquinuclidin-2-yl]methyl nicotinate* (**2d**): white solid; yield 86%; mp 149–150 °C; UV (MeOH c = 4 × 10^−5^ M) λ_max_ (log ε) 208 (4.57), 222 (4.59), 264 (3.78), 322 (3.70), 334 (3.70); IR (KBr) ν 3049 (C=C–H); 2975, 2940, 2872, 2854 (C–H_aliph_); 1716 (C=O); 1621, 1587, 1564, 1510 (C–C_arom_); 1462, 1443, 1420, 1334, 1288; 1266, 1246 (C–O); 1194, 1175, 1112, 1073, 1041, 1021, 990, 921, 855, 847, 815, 767, 739, 712, 701, 677, 643, 623, 562 cm^−1^; ^1^H NMR (DMSO–*d*6, 500 MHz) δ 1.38–1.47 (1H, m, CH_2_), 1.54 (1H, dd, *J* = 13.0, 7.7 Hz, CH_2_), 1.63–1.72 (1H, m, CH_2_), 1.73–1.78 (1H, m, CH), 2.03–2.13 (1H, m, CH_2_), 2.15–2.24 (1H, m, CH–CH=CH_2_), 2.38–2.45 (1H, m, CH_2_), 2.45–2.53 (1H, m, CH_2_), 2.83 (1H, dd, *J* = 13.4, 10.0 Hz, CH_2_), 3.11–3.23 (1H, m, CH_2_), 3.54 (1H, q, *J* = 8.5 Hz, CH–N), 3.94 (3H, s, OMe), 4.94–5.04 (2H, m, =CH_2_), 5.93–6.05 (1H, m, CH=CH_2_), 6.57 (1H, d, J = 8.8 Hz, CH–O), 7.43 (1H_quin_, dd, *J* = 9.2, 2.7 Hz), 7.56–7.61 (1H_quin_, m), 7.61–7.65 (1H_quin_+1H_Py_, m), 7.96 (1H_quin_, d, *J* = 9.2 Hz), 8.38 (1H_Py_, dt, *J* = 8.1, 2.0 Hz), 8.70 (1H_quin_, d, *J* = 4.6 Hz), 8.84 (1H_Py_, dd, *J* = 4.8, 1.7 Hz), 9.23 (1H_Py_, dd, *J* = 2.2, 0.6 Hz); ^13^C NMR (DMSO-*d*6, 125 MHz) δ 25.71 (CH_2_), 27.72 (CH), 27.89 (CH_2_), 40.10 (CH–CH=CH_2_), 42.18 (CH_2_), 56.19 (OMe), 56.41 (CH_2_), 60.40 (CH–N), 75.33 (CH–O), 102.66 (1CH_quin_), 114.82 (=CH_2_), 119.74 (1CH_quin_), 121.99 (1CH_quin_), 124.62 (1CH_Py_), 131.90 (1CH_quin_), 137.6 (1CH_Py_), 143.02 (CH=CH_2_), 148.14 (1CH_quin_), 150.76 (1CH_Py_), 154.63 (1CH_Py_), 125.77, 127.46, 144.54, 144.98, 157.85, 164.65 (6C_quater_); MS *m*/*z* (I_rel_, %) 430.30 [M+H]^+^ (100); Anal. calcd. for C_26_H_27_N_3_O_3_ (429.52): C, 72.71; H, 6.34; N, 9.78%; Found: C, 73.01; H, 6.50; N, 9.61%. 

*(R)-(6-methoxyquinolin-4-yl)[(1S,2R,4S,5R)-5-vinylquinuclidin-2-yl]methyl isonicotinate* (**2e**): white solid; yield 88%; mp 153–154 °C; UV (MeOH c = 4 × 10^−5^ M) λ_max_ (log ε) 209 (4.59), 231 (4.51), 278 (3.78), 320 (3.60), 333 (3.70); IR (KBr) ν 3071, 3033 (C=C–H); 2927, 2862 (C–H_aliph_), 1732 (C=O), 1621, 1593, 1560, 1508 (C–C_arom_); 1474, 1432, 1407, 1363, 1324; 1273, 1226 (C–O); 1116, 1063, 1029, 991, 913, 849, 830, 754, 706, 673 cm^−1^; ^1^H NMR (DMSO–*d*6, 500 MHz) δ 1.40–1.50 (1H, m, CH_2_), 1.54 (1H, dd, *J* = 13.1, 7.9 Hz, CH_2_), 1.65–1.72 (1H, m, CH_2_), 1.76–1.81 (1H, m, CH), 2.03–2.13 (1H, m, CH_2_), 2.18–2.26 (1H, m, CH–CH=CH_2_), 2.39–2.42 (1H, m, CH_2_), 2.42–2.52 (1H, m, CH_2_), 2.84 (1H, dd, *J* = 13.6, 10.1 Hz, CH_2_), 3.11–3.21 (1H, m, CH_2_), 3.53 (1H, q, *J* = 8.5 Hz, CH–N), 3.94 (3H, s, OMe), 4.97–5.07 (2H, m, =CH_2_), 5.96–6.06 (1H, m, CH=CH_2_), 6.55 (1H, d, J = 8.8 Hz, CH–O), 7.44 (1H_quin_, dd, *J* = 9.2, 2.7 Hz), 7.58–7.63 (2H_quin_, m), 7.92 (2H_Py_, m, *J* = 6.0, 1.7 Hz), 7.96 (1H_quin_, d, *J* = 9.2 Hz), 8.70 (1H_quin_, d, *J* = 4.6 Hz), 8.84 (2H_Py_, dd, *J* = 6.0, 1.6 Hz); ^13^C NMR (DMSO-*d*6, 125 MHz) δ 25.75 (CH_2_), 27.70 (CH), 27.88 (CH_2_), 40.06 (CH–CH=CH_2_), 42.18 (CH_2_), 56.24 (OMe), 56.35 (CH_2_), 60.45 (CH–N), 75.69 (CH–O), 102.65 (1CH_quin_), 114.91 (=CH_2_), 119.68 (1CH_quin_), 122.06 (1CH_quin_), 123.15 (2CH_Py_), 131.91 (1CH_quin_), 143.06 (CH=CH_2_), 148.16 (1CH_quin_), 151.54 (2CH_Py_), 127.48, 136.90, 144.51, 144.88, 157.87, 164.61 (6C_quater_); MS *m*/*z* (I_rel_, %) 430.30 [M + H]^+^ (100); Anal. calcd. for C_26_H_27_N_3_O_3_ (429.52): C, 72.71; H, 6.34; N, 9.78%; Found: C, 73.01; H, 6.50; N, 9.61%. 

*(R)-(6-methoxyquinolin-4-yl)((1S,2R,4S,5R)-5-vinylquinuclidin-2-yl)methyl (3R,5R,7R)-adamantane-1-carboxylate* (**2f**): white solid; yield 91%; mp 162–163 °C; UV (MeOH c = 5 × 10^−5^ M) λ_max_ (log ε) 232 (4.46), 279 (3.54), 320 (3.60), 333 (3.70); IR (KBr) ν 3075 (C=C–H), 2933, 2906, 2851 (C–H_aliph_); 1803, 1736 (C=O); 1621, 1591, 1508 (C–C_arom_), 1470, 1453, 1431, 1344; 1240, 1227 (C–O); 1156, 1103, 1079, 1032, 997, 973, 935, 855, 830, 717, 643 cm^−1^; ^1^H NMR (CDCl_3_, 500 MHz) δ 1.44–1.57 (2H, m, CH_2_), 1.64–1.78 (6H, m), 1.78–1.84 (1H, m), 1.84–1.89 (1H, m), 1.90–1.95 (6H, m), 1.95–2.02 (1H, m), 2.02–2.08 (3H, m), 2.23–2.31 (1H, m), 2.60–2.70 (1H, m), 3.07 (1H, dd, *J* = 13.7, 10.3 Hz), 3.10–3.17 (2H, m), 3.40–3.51 (1H, m), 3.88 (3H, s, OMe), 4.87–4.99 (2H, m, =CH_2_), 5.53 (1H, d, *J* = 4.1 Hz, CH–O), 5.68–5.77 (1H, m, CH=CH_2_), 7.21 (1H_quin_, d, *J* = 2.5 Hz), 7.29 (1H_quin_, dd, *J* = 9.2, 2.6 Hz), 7.47 (1H_quin_, d, *J* = 4.5 Hz), 7.94 (1H_quin_, d, *J* = 9.2 Hz), 8.60 (1H_quin_, d, *J* = 4.5 Hz); ^13^C NMR (CDCl_3_, 125 MHz) δ 21.99 (CH_2_), 27.77 (CH_2_), 27.84 (CH), 28.02 (CH), 36.48 (3CH_2_), 38.45 (3CH_2_), 40.08 (CH–CH=CH_2_), 43.37 (CH_2_), 55.86 (OMe), 57.12 (CH_2_), 60.13 (CH–N), 72.10 (CH–O), 101.50 (1CH_quin_), 114.59 (=CH_2_), 118.64 (1CH_quin_), 121.65 (1CH_quin_), 131.71 (1CH_quin_), 141.95 (CH–CH=CH_2_), 147.71 (1CH_quin_), 36.57, 126.80, 144.39, 147.76, 157.90, 173.56 (6C_quater_); Anal. calcd. for C_31_H_38_N_2_O_3_ (486.66): C, 76.51; H, 7.87; N, 5.76%; Found: C, 76.81; H, 7.98; N, 5.52%.

#### 3.4.2. General Procedure for the Synthesis of Compounds **3a–e**

A mixture of 0.02 mol of ester **2a–e** in 30 mL of dry dichloromethane and 0.08 mol (5 mL) of dry methyl iodide was kept for 14 days in the dark. Then, the precipitated product was filtered off, washed with 2 × 5 mL of methylene chloride and dried in a vacuum.

*(1S,2R,4S,5R)-2-(R)-[(4,5-dichloroisothiazole-3-carbonyl)oxy)(6-methoxy-1-methylquinolin-1-ium-4-yl]methyl-1-methyl-5-vinylquinuclidin-1-ium diiodide* (**3a**): orange solid; yield 98%; mp 174–175 °C; UV (MeOH c = 4 × 10^−5^ M) λ_max_ (log ε) 254 (4.48), 278 (3.70), 318 (3.60), 356 (3.70); IR (KBr) ν 3072 (C=C–H); 2997, 2925, 2855 (C–H_aliph_); 1742 (C=O), 1615, 1591, 1532 (C–C_arom_); 1475, 1460, 1440, 1430, 1415, 1378, 1350; 1274, 1243 (C–O); 1185, 1160, 1118, 1077, 1034, 1020, 1002, 970, 835, 910, 829, 795, 726, 715, 690, 514 cm^−1^; ^1^H NMR (DMSO–*d*6, 500 MHz) δ 1.70–1.81 (1H, m, CH_2_), 2.01–2.11 (1H, m, CH_2_), 2.13–2.20 (1H, m, CH), 2.22–2.30 (1H, m, CH_2_), 2.50–2.59 (1H, m, CH_2_), 2.86–2.95 (1H, m, CH–CH=CH_2_), 3.51 (3H, s, MeN), 3.42–3.62 (1H, m, CH_2_), 3.72–3.81 (1H, m, CH_2_), 3.83–3.90 (1H, m, CH–N), 4.06–4.15 (2H, m, CH_2_), 4.18 (3H, s, OMe), 4.64 (3H, s, MeN), 5.04–5.09 (1H, m, CH=CH_2_), 5.13–5,19 (1H, m, CH=CH_2_), 5.72–5.83 (1H, m, CH=CH_2_), 7.31 (1H, s, CH–O), 7.60 (1H_quin_, d, *J* = 9.2, 2.6 Hz), 8.03 (1H_quin,_ dd, *J* = 9.7, 2.5 Hz), 8.24 (1H_quin_, d, *J* = 6.1 Hz), 8.56 (1H_quin_, d, *J* = 9.7 Hz), 9.31 (1H_quin_, d, *J* = 6.2 Hz); ^13^C NMR (DMSO-*d*6, 125 MHz) δ 21.33 (CH_2_), 24.76 (CH_2_), 26.42 (CH), 37.98 (CH–CH=CH_2_), 46.40 (MeN), 49.65 (MeN), 54.91 (CH_2_), 57.06 (OMe), 65.10 (CH–N), 65.25 (CH_2_), 70.23 (CH–O), 104.30 (1CH_quin_), 117.58 (=CH_2_), 120.88 (1CH_quin_), 122.71 (1CH_quin_), 127.84 (1CH_quin_), 138.19 (CH=CH_2_), 146.92 (1CH_quin_), 125.86, 127.46, 134.66, 149.15, 151.15, 152.74, 156.94, 160.20 (8C_quater_); MS *m*/*z* (I_rel_, %) 534.10 [M + H - 2I]^+^ (6.6); Anal. calcd. for C_26_H_29_Cl_2_I_2_N_3_O_3_S (788.30): C, 39.61; H, 3.71; Cl, 8.99; I, 32.20; N, 5.33; S, 6.00%; Found: C, 39.89; H, 3.79; Cl, 8.72; I, 31.80; N, 5.05; S, 5.68%. 

*(1S,2R,4S,5R)-2-[(R)-(6-methoxy-1-methylquinolin-1-ium-4-yl)(5-phenylisoxazole-3-carbonyl)oxymethyl]-1-methyl-5-vinylquinuclidin-1-ium diiodide* (**3b**): orange solid; yield 94%; mp 190–192 °C; UV (MeOH c = 4 × 10^−5^ M) λ_max_ (log ε) 213 (4.74), 254 (4.59), 272 (4.15), 319 (3.70), 356 (3.70); IR (KBr) ν 3073 (C=C–H); 3000, 2923, 2853 (C–H_aliph_); 1750 (C=O); 1615, 1590, 1570, 1533 (C–C_arom_); 1476, 1442, 1379; 1275, 1221(C–O); 1163, 1124, 1071, 1036, 1020, 993, 947, 918, 829, 794, 762, 690 cm^−1^; ^1^H NMR (DMSO–*d*6, 500 MHz) δ 1.75–1.85 (1H, m, CH_2_), 2.07–2.16 (1H, m, CH_2_), 2.17–2.23 (1H, m, CH), 2.33–2.41 (2H, m, CH_2_), 2.51–2.60 (1H, m, CH_2_), 2.87–2.97 (1H, m, CH–CH=CH_2_), 3.45–3.53 (3H, m, NMe), 3.56 (3H, s, NMe), 3.57–3.67 (1H, m, CH_2_), 3.74–3.81 (1H, m, CH_2_), 3.86–3.94 (1H, m, CH_2_), 4.06–4.15 (2H, m, CH_2_ +CH–N), 4.19 (3H, s, OMe), 4.66 (3H, s, NMe), 5.05–5.10 (1H, m, CH=CH_2_), 5.15–5.21 (1H, m, CH=CH_2_), 5.75 (1H, s, CH–O), 5.75–5.85 (1H, m, CH=CH_2_), 7.34–7.37 (1H_quin_, m), 7.57–7.63 (4H, m, 1H_quin_+3H_Ar_), 7.70 (CH_isox_, s), 8.01–8.07 (3H, m, 1H_quin_+2H_Ar_), 8.33 (1H_quin_, d, *J* = 6.2 Hz), 8.58 (1H_quin_, d, *J* = 9.8 Hz), 9.33 (1H_quin_, d, *J* = 6.3 Hz); ^13^C NMR (DMSO-*d*6, 125 MHz) δ 21.19 (CH_2_), 24.80 (CH_2_), 26.50 (CH), 37.98 (CH–CH=CH_2_), 46.45 (NMe), 49.71 (NMe), 54.79 (CH_2_), 57.11 (OMe), 65.13 (CH–N), 65.26 (CH_2_), 70.28 (CH–O), 101.80 (CH_isox_), 104.28 (1CH_quin_), 117.62 (=CH_2_), 120.92 (1CH_quin_), 122.76 (1CH_quin_), 126.49 (2CH_Ar_), 127.85 (1CH_quin_), 130.04 (2CH_Ar_), 131.88 (1CH_Ar_), 138.16 (CH=CH_2_), 147.02 (1CH_quin_), 126.40, 127.46, 134.66, 149.09, 156.97, 158.26, 160.23, 172.15 (8C_quater_); MS *m*/*z* (I_rel_, %) 524.40 [M + H - 2I]^+^ (12.4); Anal. calcd. for C_32_H_35_I_2_N_3_O_4_ (779.45): C, 49.31; H, 4.53; I, 32.56; N, 5.39%; Found: C, 49.71; H, 4.79; I, 31.88; N, 5.16%. 

*(1S,2R,4S,5R)-2-[(R)-(6-methoxy-1-methylquinolin-1-ium-4-yl)(5-(p-tolyl)isoxazole-3-carbonyl)oxymethyl]-1-methyl-5-vinylquinuclidin-1-ium diiodide* (**3c**): orange solid; yield 95%; mp 193–194 °C; UV (MeOH c = 4 × 10^−5^ M) λ_max_ (log ε) 212 (4.73), 254 (4.58), 277 (4.23), 302 (3.70), 357 (3.70); IR (KBr) ν 3072 (C=C–H); 3000, 2946 (C–H_aliph_); 1749 (C=O); 1614, 1592, 1533, 1510 (C–C_arom_); 1477, 1443, 1378; 1275, 1223 (C–O); 1209, 1163, 1129, 1111, 1035, 1020, 993, 947, 917, 825, 794, 759, 715, 677, 500 cm^−1^; ^1^H NMR (DMSO–*d*6, 500 MHz) δ 1.73–1.84 (1H, m, CH_2_), 2.07–2.16 (1H, m, CH_2_), 2.17–2.23 (1H, m, CH), 2.32–2.39 (2H, m, CH_2_), 2.40 (3H, s, Me), 2.52–2.60 (1H, m, CH_2_), 2.87–2.97 (1H, m, CH–CH=CH_2_), 3.54 (3H, s, NMe), 3.56–3.64 (1H, m, CH_2_), 3.72–3.80 (1H, m, CH_2_), 3.84–3.93 (1H, m, CH_2_), 4.03–4.15 (2H, m, CH_2_ +CH–N), 4.18 (3H, s, OMe), 4.65 (3H, s, NMe), 5.06–5.10 (1H, m, CH=CH_2_), 5.14–5.21 (1H, m, CH=CH_2_), 5.75 (1H, s, CH–O), 5.78–5.84 (1H, m, CH=CH_2_), 7.41 (2H_Ar_, d, *J* = 8.1 Hz), 7.57–7.64 (1H_quin_), 7.61 c (CH_isox_, m), 7.92 (2H_Ar_, d, J = 8.1 Hz), 8.05 (1H_quin_, dd, *J* = 9.8, 2.5 Hz), 8.30 (1H_quin_, d, *J* = 6.1 Hz), 8.58 (1H_quin_, d, *J* = 9.7 Hz), 9.32 (1H_quin_, d, *J* = 6.2 Hz); ^13^C NMR (DMSO-*d*6, 125 MHz) δ 21.18 (CH_2_), 21.65 (Me), 24.81 (CH_2_), 26.49 (CH), 37.97 (CH–CH=CH_2_), 46.43 (NMe), 49.71 (NMe), 54.81 (CH_2_), 57.09 (OMe), 65.13 (CH–N), 65.31 (CH_2_), 70.24 (CH–O), 101.12 (CH_isox_), 104.29 (1CH_quin_), 117.62 (=CH_2_), 120.90 (1CH_quin_), 122.75 (1CH_quin_), 126.45 (2CH_Ar_), 127.84 (1CH_quin_), 130.57 (2CH_Ar_), 138.17 (CH=CH_2_), 147.04 (1CH_quin_), 123.76, 127.46, 134.67, 141.99, 149.13, 156.90, 158.31, 160.24, 172.34 (9C_quater_); Anal. calcd. for C_33_H_37_I_2_N_3_O_4_ (793.48): C, 49.95; H, 4.70; I, 31.99; N, 5.30%; Found: C, 50.21; H, 4.81; I, 31.64; N, 5.11%. 

*(1S,2R,4S,5R)-2-[(R)-(6-methoxy-1-methylquinolin-1-ium-4-yl)((1-methylpyridin-1-ium-3-carbonyl)oxy)methyl]-1-methyl-5-vinylquinuclidin-1-ium triiodide* (**3d**): reddish orange solid; yield 85%; mp 178 °C (with decomposition); UV (MeOH c = 4 × 10^−5^ M) λ_max_ (log ε) 217 (4.81), 254 (4.49), 272 (3.70), 318 (3.70), 355 (3.70); IR (KBr) ν 3075 (C=C-H), 3025, 2994, 2923, 2854 (C–C_aliph_); 1740 (C=O); 1636, 1617, 1591, 1532 (C–C_arom_); 1475, 1437, 1371, 1293; 1276, 1243 (C–O); 1214, 1165, 1098, 1034, 1020, 1000, 947, 927, 797, 727, 659 cm^−1^; ^1^H NMR (DMSO–*d*6, 500 MHz) δ 1.73–1.87 (1H, m, CH_2_), 2.02–2.13 (1H, m, CH_2_), 2.13–2.21 (1H, m, CH_2_), 2.26–2.38 (1H, m, CH), 2.67–2.78 (1H, m, CH_2_), 2.85–2.97 (1H, m, CH–CH=CH_2_), 3.32 (3H, s, NMe), 3.52–3.64 (1H, m, CH_2_), 3.75–3.84 (1H, m, CH_2_), 3.85–3.92 (1H, m, CH_2_), 3.92–4.01 (1H, m, CH_2_), 4.02–4.11 (1H, m, CH–N), 4.20 (3H, s, OMe), 4.54 (3H, s, NMe), 4.65 (3H, s, NMe), 5.05 (1H, d, =CH_2_), 5.17 (1H, d, =CH_2_), 5.76–5.85 (1H, m, CH=CH_2_), 7.37–7.44 (1H, m, CH–O), 7.65 (1H_quin_, d, *J* = 1.5 Hz), 8.05 (1H_quin_, dd, *J* = 9.7, 2.6 Hz), 8.40 (1H_quin_, dd, *J* = 8.0, 6.2 Hz), 8.47 (1H_Py_, d, *J* = 6.1 Hz), 8.57 (1H_quin_, d, *J* = 9.7 Hz), 9.16 (1H_quin_, d, *J* = 8.2 Hz), 9.30 (1H_Py_, d, *J* = 6.2 Hz), 9.37 (1H_Py_, d, *J* = 6.3 Hz), 9.67–9.71 (1H_Py_, m); ^13^C NMR (DMSO-*d*6, 125 MHz) δ 21.07 (CH_2_), 25.21 (CH_2_), 26.58 (CH), 37.97 (CH–CH=CH_2_), 46.37 (NMe), 49.25 (NMe), 49.69 (NMe), 54.74 (CH_2_), 57.17 (OMe), 64.85 (CH–N), 65.36 (CH_2_), 70.51 (CH–O), 104.48 (1CH_quin_), 117.56 (=CH_2_), 121.09 (1CH_quin_), 122.72 (1CH_quin_), 127.70 (1CH_quin_), 128.50 (1CH_Py_), 138.48 (CH=CH_2_), 146.07 (1CH_Py_), 146.82 (1CH_quin_), 147.27 (1CH_Py_), 149.87 (1CH_Py_), 127.55, 129.03, 134.60, 149.01, 160.31, 161.07 (6C_quater_); Anal. calcd. for C_29_H_36_I_3_N_3_O_3_ (855.33): C, 40.72; H, 4.24; I, 44.51; N, 4.91%; Found: C, 40.96; H, 4.51; I, 44.11; N, 4.75%. 

*(1S,2R,4S,5R)-2-[(R)-(6-methoxy-1-methylquinolin-1-ium-4-yl)((1-methylpyridin-1-ium-4-carbonyl)oxy)methyl]-1-methyl-5-vinylquinuclidin-1-ium triiodide* (**3e**): reddish orange solid; yield 83%; mp 200 °C (with decomposition); UV (MeOH c = 4 × 10^−5^ M) λ_max_ (log ε) 219 (4.81), 254 (4.46), 279 (3.70), 318 (3.70), 355 (3.70); IR (KBr) ν 3075 (C=C–H); 3001, 2924, 2854 (C–H_aliph_); 1745 (C=O); 1639, 1616, 1592, 1531 (C–C_arom_); 1450, 1432, 1376; 1270, 1244 (C–O); 1218, 1162, 1001, 1035, 1019, 1000, 943, 917, 900, 824, 723, 671 cm^−1^; ^1^H NMR (DMSO–*d*6, 500 MHz) δ 1.77–1.87 (1H, m, CH_2_), 2.02–2.14 (1H, m, CH_2_), 2.15–2.22 (1H, m, CH_2_), 2.24–2.34 (1H, m, CH), 2.64–2.72 (1H, m, CH_2_), 2.87–2.96 (1H, m, CH–CH=CH_2_), 3.33 (3H, s, NMe), 3.52–3.64 (1H, m, CH_2_), 3.76–3.84 (1H, m, CH_2_), 3.85–3.94 (1H, m, CH_2_), 3.95–4.03 (1H, m, CH_2_), 4.04–4.14 (1H, m, CH–N), 4.20 (3H, s, OMe), 4.54 (3H, s, NMe), 4.65 (3H, s, NMe), 5.03–5.08 (1H, m, =CH_2_), 5.14–5.21 (1H, m, =CH_2_), 5.77–5.86 (1H, m, CH=CH_2_), 7.29–7.36 (1H, m, CH–O), 7.62 (1H_quin_, d, *J* = 1.5 Hz), 8.05 (1H_quin_, dd, *J* = 9.7, 2.6 Hz), 8.43 (1H_quin_, d, *J* = 6.2 Hz), 8.57 (1H_quin_, d, *J* = 9.8 Hz), 8.71–8.76 (2H_Py_, m), 9.30 (2H_Py_, d, *J* = 6.7 Hz), 9.38 (1H_Py_, d, *J* = 6.4 Hz); ^13^C NMR (DMSO-*d*6, 125 MHz) δ 20.98 (CH_2_), 25.09 (CH_2_), 26.54 (CH), 37.92 (CH–CH=CH_2_), 46.38 (NMe), 49.31 (NMe), 49.63 (NMe), 54.39 (CH_2_), 57.16 (OMe), 64.91 (CH–N), 65.29 (CH_2_), 70.77 (CH–O), 104.37 (1CH_quin_), 117.60 (=CH_2_), 120.96 (1CH_quin_), 122.73 (1CH_quin_), 127.72 (1CH_quin_), 127.84 (2CH_Py_), 138.36 (CH=CH_2_), 146.91 (1CH_quin_), 147.55 (2CH_Py_), 127.53, 134.58, 143.11, 148.91, 160.27, 161.61 (6C_quater_); Anal. calcd. for C_29_H_36_I_3_N_3_O_3_ (855.33): C, 40.72; H, 4.24; I, 44.51; N, 4.91%; Found: C, 40.96; H, 4.51; I, 44.11; N, 4.75%. 

## 4. Conclusions

A convenient method for the preparation of natural alkaloid quinine derivatives via an acylation reaction with 4,5-dichloroisothiazole-3-, 5-arylisoxazole-3-, adamantane- and hydrochlorides of pyridine-3- and pyridine-4-carbonyl chlorides has been developed. According to bioassay data, all tested substances did not show pronounced antiviral properties in the range of doses studied, except for compounds **2e**, **3b**, **3c** and **3e**, which showed more pronounced antiviral properties. However, the presence of antimicrobial and analgesic activity in newly synthesized compounds, particularly compounds **2b**, **2e**, **2f**, **3b**, **3c** and **3e** with isoxazole, pyridine and adamantane fragments, makes it possible to consider them promising for further study of their pharmacological properties.

## Data Availability

The data presented in this study are available in the article or Appendix A.

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
