# Peer review of "Quinine Esters with 1,2-Azole, Pyridine and Adamantane Fragments"

_molecules, 2022, doi:10.3390/molecules27113476_

Round 1

Reviewer 1 Report

The manuscript reports the preparation of quinine esters incorporating substituted thiazole, arylisoxazole and pyridine heterocycles as well as adamantine system.  In my opinion, the manuscript needs revision and modifications before consideration for publication: -

  • The rational for the synthesis of compounds is unclear which make the main objectives need to be outlined and presented clearly. From medicinal chemistry point of view, no rational presented for the synthesis of compounds. In addition, how did authors choose such biological applications. On what basis the selected viruses and organisms were selected.? From organic synthesis point of view, authors may pay attention in the introduction part for the importance and wide synthetic applications of esters (which is one of the main objectives for the special issue subject).
  • Please discuss in more details the effect of using the hydrochloride form of the acetylating agents in case of nicotinyl and isonicotinyl derivatives on increasing the yield, possible interpretation and advantages as well as any factors played any role in improving the yields.
  • The antimicrobial part is too brief, please discuss in details. I addition please, provide the results of a reference drug for the antimicrobial activity for accounting clearly of the degree of activity of the synthesized compounds
  • Please, discuss the SAR and the possible effects of structural characteristics on the observed biological activity results
  • Please, mention the composition ratio of the mixed solvent of crystallization and similarly, please, write the amount in gm of trimethylamine in the experiment in line 294
  • Please unify the description form of the IR spectral data for all values (providing the functional group type for all or not for all)
  • Please, check carefully the splitting description of the assigned hydrogens in H NMR, most of the methylene hydrogens are expected to appear as “dd” as a result of the chirality centers
  • Please unify the description form of the hydrogens in the spectral data part for H NMR
  • Please, revise carefully for misprint typos such as bold format for compound numbers, decimal point in numbers, subscript and superscript numbers and characters, and spaces especially in the experimental part
  • Authors mentioned that there is are NMR spectra in the supplementary materials file, I did not find any NMR chart, instead 2 different IR charts for each compound!!!

Reviewer 2 Report

  1. “Conjugate” is not the best term to describe the compounds obtained by the authors
  2. The authors should provide some additional information and/or their considerations about the ester group aimed to bind quinine with heterocyclic moieties. There are several references in the Introduction but as the aim of the work is to combine chinine with heterocycles via this linker to modify biological activity, more attention is to be drawn to this approach. 
  3. Keywords and line 63: “adamantine” should be “adamantane” 3.
  4. Table 5, last column heading. It is not clear why the authors put here “% Relative number of writhing to control” In fact, the figures show percentage by which the number of writhing decreased. This is properly indicated in the text above this table. Why did the authors used 25 mg/kg dose for their compounds and 8 mg/kg for sodium diclofenac?
  5. Why virus-inhibiting activity was studied only for 4 selected compounds?

Reviewer 3 Report

  1. A limited number of molecules is shown in the manuscript. It is not clear why the authors investigated just six groups at 9-position. The authors should better explain the rationale. Is there any idea about the target? This could move the design better.
  2. Why are these molecules not evaluated as antiplasmodial agents? Why was the screening on C.albicans and other infective agents? The authors should explain better the choice.
  3. Line 98, line 109 (acute toxicity on macrophage): the authors said "concentration range of 0.03 - 0.2%, " but the concentration tested is not so clear.
  4.  Some compounds are not tested. For example, just 4 compounds were tested against H3N2, 4  compounds against H7N1,  etc.  Why?
  5. Reference drugs were not used in antiviral antifungal and antibacterial tests. They should be used and their activity compared to those of tested cmps. 
  6. Can it be assumed that salts 3a-e are able to cross the membrane and reach their target?
  7. Why was 3f not prepared?
  8. Scheme 1:  it not clear the meaning of X and R1 for compounds 2c, 3c
  9. In table 4, MIC value should be expressed in (micro)molarity
  10. In material and methods information about UV instruments are missing.
  11. Par. 3.4.1: the synthesis of 2f is missing

Minor comments

Line 87: 1H and 13C must have the number as apix.

In the whole text the authors frequently use the terms ``samples" and "drugs''. In my opinion, "molecules" or "compounds" or "quinine derivatives" could be more appropriate. Moreover, the term drug is not appropriate in this preliminary phase of the study.

The term "in vivo" and "in vitro" must be italicized.

The unit "l" (liter) must be "L" (whole text)

Round 2

Reviewer 1 Report

In the revised version, authors response to the major aspects and required changes were performed. It can be accepted in its current form. 

Author Response

Dear reviewer! We have made the changes to the introduction and tried to give a clear rational for the synthesis of compounds.

Reviewer 3 Report

The manuscript was improved and most of the issues have been addressed. It is not clear the answer to my first point ("Point 1: A limited number of molecules is shown in the manuscript. It is not clear why the authors investigated just six groups at 9-position. The authors should better explain the rationale. Is there any idea about the target? This could move the design better."). It seems to me that the compounds were already available in their in-house library, instead of being prepared "ad hoc" for the present study.

I would add a minor comment: in the tables, the values at the first rows should be with dot, not with comma (0,003 --> 0.003, and so on).

Author Response

Dear reviewer! We have made the changes to the introduction and tried to give a clear rational for the synthesis of compounds. Minor mistakes were corrected.